# Handling Long-tailed Feature Distribution in AdderNets

**Minjing Dong**[1,2], **Yunhe Wang**[2]*, **Xinghao Chen**[2], **Chang Xu**[1]
[1]School of Computer Science, University of Sydney
[2]Huawei Noah's Ark Lab
`mdon0736@uni.sydney.edu.au, yunhe.wang@huawei.com,`
`xinghao.chen@huawei.com, c.xu@sydney.edu.au`

## Abstract

Adder neural networks (ANNs) are designed for low energy cost which replace expensive multiplications in convolutional neural networks (CNNs) with cheaper additions to yield energy-efficient neural networks and hardware accelerations. Although ANNs achieve satisfactory efficiency, there exist gaps between ANNs and CNNs where the accuracy of ANNs can hardly be compared to CNNs without the assistance of other training tricks, such as knowledge distillation. The inherent discrepancy lies in the similarity measurement between filters and features, however how to alleviate this difference remains unexplored. To locate the potential problem of ANNs, we focus on the property difference due to similarity measurement. We demonstrate that unordered heavy tails in ANNs could be the key component which prevents ANNs from achieving superior classification performance since fatter tails tend to overlap in feature space. Through pre-defining Multivariate Skew Laplace distributions and embedding feature distributions into the loss function, ANN features can be fully controlled and designed for various properties. We further present a novel method for tackling existing heavy tails in ANNs with only a modification of classifier where ANN features are clustered with their tails well-formulated through proposed angle-based constraint on the distribution parameters to encourage high diversity of tails. Experiments conducted on several benchmarks and comparison with other distributions demonstrate the effectiveness of proposed approach for boosting the performance of ANNs.

## 1 Introduction

Deep Convolutional Neural Networks have been widely adopted in various computer vision tasks due to their satisfactory performance, including image classification [12, 9, 10, 20], object detection [13, 17, 6, 7], super-resolution [29, 5, 11, 18], *etc.*. However, the success of CNNs can hardly be presented for practical usage without further modifications since the majority of computer vision tasks are usually deployed on low-power platforms, such as portable and embedded devices, where the computational resources are significantly constrained while prompt inference is required. As a result, the acceleration of deep neural networks and reduction of energy cost become urgent requirements and attract massive research efforts in recent years [8, 24, 15, 27, 4, 30, 21].

Recently, Chen *et al.*[3] introduced Adder Neural Network to replace the cross-correlation in CNNs with $\ell_1$-norm for the similarity measurement between input features and filters, which eliminates the massive multiplications in CNNs. Through replacing existing multiplications with additions in deep neural networks, ANNs achieve considerable energy reduction since addition is a cheaper operation than multiplications [19, 23]. Furthermore, ANN can be more friendly for hardware designs of deep learning acceleration [25, 28]. Although ANNs achieve comparable results with CNNs, there exist

---

*Corresponding author.

35th Conference on Neural Information Processing Systems (NeurIPS 2021).

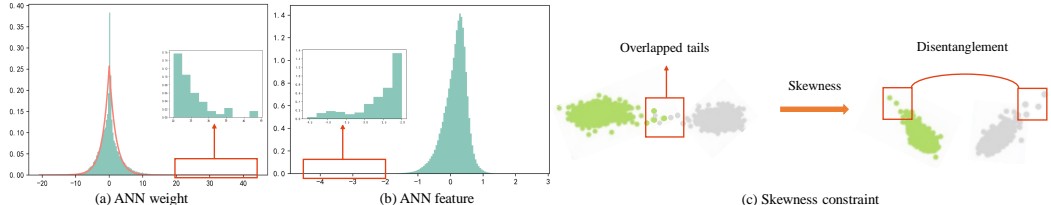

Figure 1: (a) and (b) denote histograms of ANN weights and features respectively. (c) denotes a toy example of overlapped heavy tails of feature distributions and a solution through skewness constraint.

gaps in classification task (*e.g.* 67.0% top-1 accuracy on ImageNet with ANN-18 while 69.8% with CNN-18). Progressive Kernel Based Knowledge Distillation (PKKD) has been proposed to establish ANN with better performance. However a concomitant CNN with the same architecture and a parallel optimization of both ANN and CNN are required, which largely raises the cost of training [26]. The natural difference between CNNs and ANNs mainly lies in the similarity measurement, which is always ignored and how to relieve the gaps from inherent properties of ANNs remains unexplored. Thus, the difference caused by the replacement of cosine distance with $\ell_1$ norm is shed light on to explore potentially more powerful classification capability of ANNs.

The discrepancy between ANNs and CNNs due to different similarity measurements lies in various aspects. In this work, we mainly focus on the ANN features for classification. A well-studied property of ANNs is that the features corresponding to different classes are clustered towards different centers since $\ell_1$ norm is used for similarity measurement [3]. Meanwhile, ANN weights tend to follow a Laplace distribution which denotes the prior of $\ell_1$. Actually the heavy-tailed distributions widely exist not only in ANN weights but also in ANN features, as shown in Figure1 (a) and (b). Since the features are clustered for different classes, the existing fatter tails can be easily overlapped with each other, which could hurt the classification performance.

In this paper, we propose to pre-define the feature distributions in order to model the heavy-tailedness in ANNs. We demonstrate the limitations of Gaussian for ANNs and instead make use of a mixture of Multivariate Skew Laplace Distributions which not only involves mean and variance for optimization but also includes higher-order moment skewness. With skewness, the overlapped areas of heavy tails can be reduced for preventing the entanglement of feature distributions. We propose to embed this mixture of skew Laplace into the loss function through substituting the distribution parameters for the classifier head. A likelihood regularization comes naturally for fitting ANN features to pre-defined distributions. With distribution parameters, we introduce a well-formulated angle-based constraint on the feature distributions based on their locations, covariance and skewness, which drives the distribution tails to different angle regions for disentanglement. Our proposed method improves the classification accuracy by 0.7% on both CIFAR-100 with ResNet-20 and ImageNet with ResNet-18 compared to vanilla ANN with only a modification to the ANN classifier head.

## 2  Preliminaries

**Adder Neural Networks.**   Chen *et al.*[3] proposed the Adder Neural Network to eliminate the multiplications in traditional convolution networks and replace them with additions to significantly reduce computational and energy costs. Consider an intermediate feature map $X \in \mathbb{R}^{H \times W \times c_{in}}$ in deep neural network with weight $W \in \mathbb{R}^{w \times w \times c_{in} \times c_{out}}$ where $H, W$ denote the height and width of input feature, $w$ denotes the kernel size, and $c_{in}, c_{out}$ denote the number of input and output channel respectively. The convolution and adder operation are defined as

$$Y_{conv}(m,n,c) = \sum_{i=1}^{w}\sum_{j=1}^{w}\sum_{k=1}^{c_{in}} X(m+i, n+j, k) \times W(i,j,k,c),$$

$$Y_{adder}(m,n,c) = -\sum_{i=1}^{w}\sum_{j=1}^{w}\sum_{k=1}^{c_{in}} |X(m+i, n+j, k) - W(i,j,k,c)|. \tag{1}$$

Comparing the adder operation with the traditional convolution in Eq. 1, the dot product is replaced by the $\ell_1$-norm for measuring the similarity between the filter and input feature. Although ANN can achieve similar performance to CNN, there still exist gaps between ANNs and CNNs. For example,

ANN has 0.7% accuracy drop with ResNet-32 on CIFAR-100 compared to CNN. Since the major difference between ANN and CNN lies in the operation, we mainly focus on addressing the potential adverse properties of this operation substitution for improving ANN performance.

## 3 Skew Laplace Mixture Loss with Angle-based Constraint

In this section, we analyze potential problems of ANNs and propose a novel framework which replaces classifier head with appropriate trainable distribution parameters. Furthermore, we introduce an angle-based constraint for controlling feature distribution tails to avoid potential entanglement.

### 3.1 Embedding Skew Laplace Mixture into Loss Function

With $\ell_1$-norm as similarity measurement, the weights in ANNs are close to Laplace distribution since the prior of $\ell_1$-norm is Laplace distribution, as discussed in [3]. Laplace density is expressed by the absolute difference from mean while Gaussian density is expressed by the squared difference, which results in a fatter tails of Laplace distributions compared to Gaussian distributions. We empirically verifies the heavy tails in both ANN weights and features through histograms of sampled layer from a pre-trained ANN with ResNet32 on CIFAR-10, as shown in Figure 1 (a) and (b). Another important property lies in the features for classification. As discussed and verified in [3], ANN features are clustered and classified by Manhattan distance while CNN features by cosine distance. Combing aforementioned properties, a concern arises that the heavy tails could become a potentially troublesome issue for $\ell_1$-norm based clustering classification since the feature tails corresponding to different classes are more likely to be overlapped, which significantly reduces the classification margin and constrains the generalization of ANNs, as illustrated in Figure 1 (c). Directly tackling this issue can be rather difficult since the distributions of features are unknown, which motivates us to pre-define an appropriate feature distribution for ANNs to obtain a high-level control of heavy tails.

Thus, we make an assumption that deep features of neural networks follow learnable distributions, with which the feature distribution can be better formed to achieve expected properties. An naive selection can be Multivariate Gaussian distribution $\mathcal{N}(\mu, \Sigma)$ where $\mu$ denotes the mean and $\Sigma$ denotes the covariance matrix. However, there exist several concerns in ANNs. For example, the features and weights of ANN always form long-tailed distributions instead of the bell curves with Gaussian distributions, which indicates that pre-defined Gaussian distributions might not fit the real ANN features well. Moreover, Gaussian distribution contains two parameters $\mu$ and $\Sigma$ for optimization. However, the heavy-tailedness can hardly be controlled by these two lower-order moments, which indicates a higher-order moment is required to be involved in pre-defined distribution. Thus, Multivariate Gaussian distribution needs to be replaced by a more appropriate one in ANNs.

In this work, we make use of Multivariate Skew Laplace distribution (SL) [1] to fulfill aforementioned requirements. Considering a classification task with $K$ classes, the density of ANN last-layer features $x \in \mathbb{R}^d$ with respect to its corresponding class $k$ is given by

$$f_{SL}(x; \mu_k, \Sigma_k, \gamma_k) = \frac{|\Sigma_k|^{-1/2}}{2^d \pi^{(d-1)/2} \alpha_k \Gamma(\frac{d+1}{2})} e^{-\alpha_k \sqrt{(x-\mu_k)^T \Sigma_k^{-1}(x-\mu_k)} + (x-\mu_k)^T \Sigma_k^{-1} \gamma_k}, \quad (2)$$

where $\mu$, $\Sigma$, $\gamma$ denote the location, covariance and skewness parameters respectively, $\alpha_k = \sqrt{1 + \gamma_k^T \Sigma_k^{-1} \gamma_k}$ and $\Gamma$ denotes the Gamma function. Skew Laplace distribution is selected for several reasons. First, Laplace distribution fits ANN well since ANN filters empirically follow Laplace distributions and the heavy tails of ANN features can be well-approximated by Laplace distributions. Moreover, the skewness parameters in Multivariate Skew Laplace naturally incorporate a high-level control of distribution tails in ANNs. According to the definition of skewness, it indicates the direction and relative magnitude of a distribution deviation from its center location. Through optimizing skewness parameters to appropriate values, the overlapped area of heavy tails can be easily eliminated since skewness directly controls the shape of distributions, especially the direction of heavy tails, to achieve disentanglement in angle space, as shown in Figure 1 (c).

Note that each class $k$ follows independent $\mathcal{SL}(\mu_k, \Sigma_k, \gamma_k)$, which makes ANN feature $x$ follows a mixture of Multivariate Skew Laplace distributions. The distribution of $x$ can be computed as

$$p(x) = \sum_{k=1}^{K} \mathcal{SL}(x; \mu_k, \Sigma_k, \gamma_k) p(k) \quad (3)$$

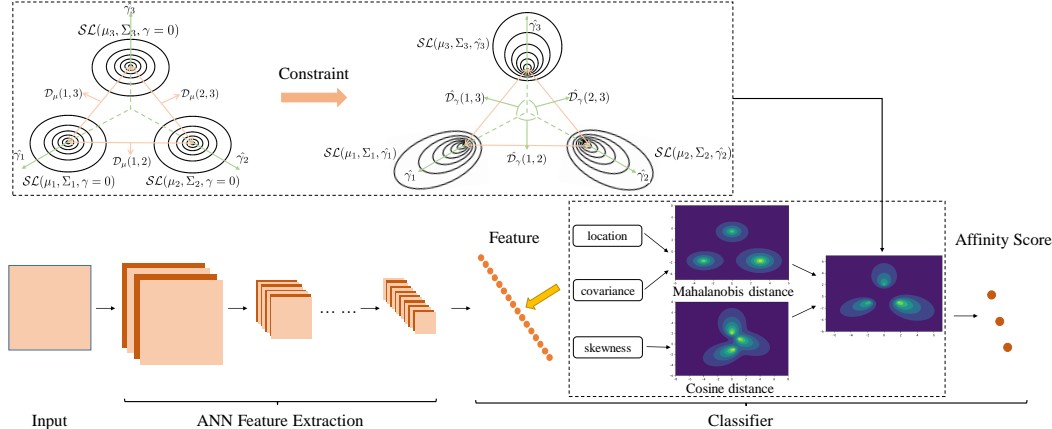

Figure 2: An illustration of proposed method. ANN feature extraction network remains the same while proposed classifier imposes constraints on features which involves both Mahalanobis and cosine distance to form a clustering classification with distribution tails dispersed in different directions.

where $p(k)$ denotes the prior probability. Although the real distribution of ANN feature is hard to derive, how well the extracted features fit this mixture of Multivariate Skew Laplace distributions can be measured through sampling subsets from training set and computing the negative log likelihood as

$$
\begin{aligned}
\mathcal{L}_{nll} = -log[\mathcal{L}(\mu, \Sigma, \gamma|x)] &= -log[f_{SL}(x|\mu_k, \Sigma_k, \gamma_k)p(k)], \\
&= \sum_{i=1}^{N} \alpha_k + 0.5|\Sigma_k| + \alpha_k \sqrt{(x_i - \mu_k)^T \Sigma_k^{-1}(x_i - \mu_k)} \\
&\quad - (x_i - \mu_k)^T \Sigma_k^{-1} \gamma_k - log[p(k)],
\end{aligned}
\tag{4}
$$

where $\mathcal{L}$ denotes the likelihood. Through incorporating the negative log likelihood into the objective loss, ANN features $x$ can be well-formulated as learned $\mathcal{SL}(\mu_k, \Sigma_k, \gamma_k)$. Since ANN features now follow a mixture of Multivariate Skew Laplace distributions, the classification loss can be easily derived. With the predefined distribution with its parameters, we can derive the conditional probability of class label $y_i \in [1, K]$ through Bayes' theorem. Similar to the traditional classification loss which computes the cross-entropy between affinity scores and one-hot encoding of class labels, the classification loss can be computed through treating $p(y_i|x_i)$ as the affinity score as

$$
\begin{aligned}
\mathcal{L}_c = -\frac{1}{N} \sum_{i=1}^{N} log[p(y_i|x_i)] &= -\frac{1}{N} \sum_{i=1}^{N} log[\frac{p(x_i|y_i)p(y_i)}{p(x_i)}], \\
&= -\frac{1}{N} \sum_{i=1}^{N} log[\frac{f_{SL}(x_i; \mu_{y_i}, \Sigma_{y_i}, \gamma_{y_i})p(y_i)}{\sum_{k=1}^{K} f_{SL}(x_i; \mu_k, \Sigma_k, \gamma_k)p(k)}].
\end{aligned}
\tag{5}
$$

Through combing Eq. 4 and 5, we embed the feature distributions into the loss function and make $x$ follow a mixture of SL distributions, where the observed heavy tails can be parameterized and a direct connection between ANN features and trainable distribution parameters can be established to provide a focus for tackling existing overlapping issue.

## 3.2 Angle-based Constraint

Note that the prior probability $p(k) = \frac{1}{K}$ and the distribution parameters of different classes are independent from each other, which indicates there is no direct correlation among different feature distributions. If we directly optimize $\mathcal{SL}(\mu_k, \Sigma_k, \gamma_k)$ for each class, it could be difficult to reduce the overlapping of distribution tails since each class is formulated independently in Sec 3.1 without taking feature distributions of other classes into consideration. Thus, we take the feature distributions of all the classes as a whole and propose to explore the constraints among classes which diverge the heavy tails of their feature distributions to enlarge classification margin. Since the type and corresponding parameters of feature distributions are available during training phase, constraints can be easily applied on the distribution parameters $\mu$, $\Sigma$ and $\gamma$ to achieve desired performance.

We now introduce a simple yet effective constraint on the feature distributions, which leads to diversity of distribution tails. We first explore the existing similarity measurements in proposed classifier. The location $\mu$ and covariance $\Sigma$ form a Mahalanobis distance $\sqrt{(x - \mu_k)^T \Sigma_k^{-1}(x - \mu_k)}$ in Eq. 2 for measuring the distance from feature $x$ to the center of class $k$. Similarly, the involvement of skewness $\gamma$ forms inner product $(x - \mu_k)^T \Sigma_k^{-1} \gamma_k$ in Eq. 2 for measuring the cosine distance from normalized feature $x$ to the skewness of class $k$. Now we focus on the similarity measurement among classes. Given random pair of classes $m$ and $n$, the relative distance from class $m$ to $n$ can be easily derived through replacing feature $x$ with the distribution parameters of other classes in aforementioned distance. Since our objective is incorporating disentanglement of feature distributions, we propose to arrange their heavy tails to different angle regions. In order to fulfill this potential, we measure the distance of feature distribution skewness among different classes and impose constraint to encourage tail divergence in angle space. Given a set of skewness parameters $\gamma_{1:K} = [\gamma_1, \gamma_2, ..., \gamma_K] \in \mathbb{R}^{K \times d}$, we compute the skewness distance $\mathcal{D}_\gamma \in \mathbb{R}^{K \times K}$ through computing the cosine similarity between each class pair. For example, $\mathcal{D}_\gamma$ between class $m$ and $n$ is given by

$$\mathcal{D}_\gamma(m, n) = cos\theta(\gamma_m, \gamma_n) = \frac{\gamma_m \cdot \gamma_n}{\|\gamma_m\| \, \|\gamma_n\|}. \tag{6}$$

Although $\mathcal{D}_\gamma$ can be easily derived, the target $\hat{\mathcal{D}}_\gamma$ remains undefined. Note that the skewness itself denotes the direction and relative magnitude of its corresponding distribution heavy-tailedness, which controls the shape of distribution and provides an implicit similarity measurement among classes, as illustrated in Figure 2. Meanwhile, the first-order moments $\mu$ provides a more explicit measurement for describing the distance among classes, which suggests the distances among the centers of different clusters could become an indicator for obtaining target skewness distance among classes. Thus, we propose to make use of location parameters $\mu$ to obtain an adaptive target $\hat{\mathcal{D}}_\gamma$. For class $m$, we first take the location parameter $\mu_m$ as the center point and make use of square of Mahalanobis distance for measuring the location difference between $\mu_u$ and the feature distributions of other classes as

$$\mathcal{D}_\mu(m, n) = D^2_{\mathcal{SL}_n}(\mu_m) = (\mu_m - \mu_n)^T \Sigma_n^{-1}(\mu_m - \mu_n), \tag{7}$$

where $\mathcal{SL}_n$ denotes the Multivariate Skew Laplace distribution corresponding to class $n$ and $D$ denotes the Mahalanobis distance. With Eq. 6 and 7, we design the constraint based on two simple principles: (a). For class $m$, target $\hat{\mathcal{D}}_\gamma(m, n)$ where $n \in [1, 2, ..., K]$ can always achieve the maximum and minimum values of cosine distance to fully utilize the angle space; (b). $\mathcal{D}_\mu(m, n)$ determine how $\hat{\mathcal{D}}_\gamma(m, n)$ are distributed in range $[-1, 1]$ where the tail direction difference $\hat{\mathcal{D}}_\gamma$ is inverse proportion to the location difference $\mathcal{D}_\mu$. Thus, the propose angle-based constraint is given by

$$\mathcal{L}_{ac} = \frac{1}{K^2} \sum_{m=1}^{K} \sum_{n=1}^{K} [\mathcal{D}_\gamma(m, n) - \hat{\mathcal{D}}_\gamma(m, n)]^2,$$

$$\text{where } \hat{\mathcal{D}}_\gamma(m, n) = \frac{max(\mathcal{D}_\mu(m, 1{:}K)) - 2\,\mathcal{D}_\mu(m, n)}{max(\mathcal{D}_\mu(m, 1{:}K))}, \tag{8}$$

where function $max$ takes the maximum value among all $K$ classes. Eq. 8 computes the mean squared error between current skewness distance and target one, where $\hat{\mathcal{D}}_\gamma$ purely depends on current location and covariance parameters of feature distributions to form an adaptive target distance. $\hat{\mathcal{D}}_\gamma$ is designed by projecting the pre-computed Mahalanobis distance into range $[-1, 1]$ in a reverse order. Note that $\hat{\mathcal{D}}_\gamma(m, m)$ becomes 1 since $\mathcal{D}_\mu(m, m) = 0$ and $\hat{\mathcal{D}}_\gamma(m, n)$ becomes $-1$ when class $n$ has the maximum Mahalanobis distance $\mathcal{D}_\mu(m, n) = max(\mathcal{D}_\mu(m, 1{:}K))$. An illustration of how proposed constraint influence the pre-defined feature distributions is shown in Figure 2. With the combination of Mahalanobis and cosine distance for angle-based constraint, Skew Laplace distribution introduces a clustering classification without long tails entanglement through controlling the heavy tails. Finally, combing Eq. 4, Eq. 5 and Eq. 8, the objective loss is defined as

$$\mathcal{L}_{obj} = \mathcal{L}_c + \lambda \mathcal{L}_{nll} + \beta \mathcal{L}_{ac}, \tag{9}$$

where $\lambda$ and $\beta$ are the hyper-parameters for balancing these terms.

### 3.3 Optimization

Our final objective loss named Skew Laplace Loss with angle-based constraint (SLAC) can be simply optimized through stochastic gradient decent (SGD). Similar to vanilla ANN, we make use

---

**Algorithm 1** Skew Laplace Mixture Loss with Angle-based Constraint for AdderNet

---

**Input:** The training set $\{\mathcal{X}, \mathcal{Y}\}$; Batch size $n$; Hyperparameter $\lambda$, $\beta$;
Initialize ANN network $\mathcal{N}$ with $W$ where $W$ is initialized by uniform distribution;
Initialize Multivariate Skew Laplace distributions $SL(\mu, \Sigma, \gamma)$ where $\mu$ is initialized by Xavier initialization, $\Sigma$ by 1.0 and $\gamma$ by 0.0;
**while** not converge **do**
    Sample a batch of data $\{\mathcal{X}, \mathcal{Y}\}_{i=1}^{n}$ from $\{\mathcal{X}, \mathcal{Y}\}$;
    Forward ANN network to get feature $x_{i=1}^{n} = \mathcal{N}(\mathcal{X}_{i=1}^{n}, W)$;
    Forward classifier to get affinity score $p(\mathcal{Y}_{i=1}^{n}|x_{i=1}^{n})$ with Eq. 5;
    Calculate objective loss $\mathcal{L}_{obj}$ with Eq. 9;
    Update the $SL(\mu, \Sigma, \gamma)$ through SGD and pass gradients to $\mathcal{N}$ with Eq. 10;
    Update ANN parameters $W$ through full-precision gradient with SGD;
**end while**

---

of full-precision gradient to update filters $W$, which is computed based on $\ell_2$-norm. The major difference lies in the gradients passed by the classifier. In SLAC ANN, the partial derivative of $\mathcal{L}_{obj}$ with respect to last-layer feature $x$ is calculated as

$$
\begin{aligned}
\frac{\partial \mathcal{L}_{obj}}{\partial x} &= (1 - p(y_i|x) + \lambda)(\alpha_{y_i}((x - \mu_{y_i})^T \Sigma_{y_i}^{-1}(x - \mu_{y_i}))^{-1/2} \Sigma_{y_i}^{-1}(x - \mu_{y_i}) + \Sigma_{y_i}^{-1} \gamma_{y_i}) \\
&+ \sum_{k \neq y_i} p(k|x)(\Sigma_k^{-1} \gamma_k - \alpha_k((x - \mu_k)^T \Sigma_k^{-1}(x - \mu_k))^{-1/2} \Sigma_k^{-1}(x - \mu_k)).
\end{aligned}
\tag{10}
$$

Although proposed constraint $\mathcal{L}_{ac}$ is not directly involved in Eq. 10, the gradients passed to ANN layers from classifier are heavily determined by the feature distribution parameters $\mu$, $\Sigma$ and $\gamma$, which are significantly influenced by $\mathcal{L}_{ac}$. Similar to Eq. 10, the partial derivative of $\mathcal{L}_{obj}$ with respect to $\mu$, $\Sigma$ and $\gamma$ can be computed through SGD accordingly. The entire algorithm named Skew Laplace Mixture Loss with Angle-based constraint for AdderNet (SLAC-ANN) is shown in Algorithm 1. Since our proposed SLAC-ANN involves a constraint on the distribution parameters to encourage tail diversity in angle space which significantly improves the classification margin, the complexity of optimization is relatively enlarged .The superiority of our method becomes more obvious when the number of epochs are enlarged, which will be verified in empirical evaluations.

## 4 Experiments

In this section, we conduct empirical evaluation of the proposed SLAC ANN on several image classification benchmarks, including CIFAR-10, CIFAR-100 and ImageNet. ANN variants comparison and ablation studies are provided to demonstrate the effectiveness of the proposed algorithm.

### 4.1 Experiments on CIFAR

CIFAR-10 and CIFAR-100 dataset contain $50K$ training images and $10K$ validation images with size of $32 \times 32$ from 10 categories. All the training and validation sets are pre-processed according to the same protocol as in [9]. We make use of SGD optimizer with an initial learning rate of 0.1, weight decay of $5 \times 10^{-4}$, momentum of 0.9 and a cosine learning rate schedule. The entire training takes 800 epochs with a batch size of 256. The learning rate of trainable parameter $\Sigma$ is downscale by $1 \times 10^2$, $\lambda$ is set to 0.01 and $\beta$ to 0.1. For comparison, we include CNN, ANN and binary neural network (BNN) [30] as baselines. Note that the first and last layers are set to full-precision convolutional layer in vanilla ANN [3]. For a fair comparison, SLAC-ANN adopts a similar setting, which uses convolutional operation for the first layer but replaces the last layer by the proposed $SL$ distributions parameters. All the baseline results are cited from [3].

The classification results are reported in Table 1. We first evaluate the performance of proposed SLAC-ANN on VGG-small model [2]. SLAC-ANN achieves better performance on both CIFAR-10 and CIFAR-100 compared to other baselines. For example, SLAC-ANN improves vanilla ANN by $0.99\%$ [$72.64\% \rightarrow 73.63\%$] on CIFAR-100. We further compare on the widely used ResNet models [9]. For ResNet-20, SLAC-ANN achieves $92.29\%$ and $68.31\%$ accuracy on CIFAR-10 and CIFAR-100 respectively, which improves vanilla ANN by $0.45\%$ on CIFAR-10 and $0.71\%$ on CIFAR-100. Although higher efficiency can be achieved, BNN cannot achieve competitive classification

Table 1: Classification results on CIFAR-10 and CIFAR-100 datasets.

| Model | Method | #Mul. | #Add. | #XNOR. | CIFAR-10 | CIFAR-100 |
|---|---|---|---|---|---|---|
| VGG-small | CNN | 0.65G | 0.65G | 0 | 93.80% | 72.73% |
| | BNN | 0.05G | 0.65G | 0.60G | 89.80% | 67.24% |
| | ANN | 0.05G | 1.25G | 0 | 93.72% | 72.64% |
| | SLAC-ANN | 0.05G | 1.25G | 0 | **93.96%** | **73.63%** |
| ResNet-20 | CNN | 41.17M | 41.17M | 0 | 92.25% | 68.14% |
| | BNN | 0.45M | 41.17M | 40.72M | 84.87% | 54.14% |
| | ANN | 0.45M | 81.89M | 0 | 91.84% | 67.60% |
| | SLAC-ANN | 0.46M | 81.90M | 0 | **92.29%** | **68.31%** |
| ResNet-32 | CNN | 69.12M | 69.12M | 0 | 93.29% | 69.74% |
| | BNN | 0.45M | 69.12M | 68.67M | 86.74% | 56.21% |
| | ANN | 0.45M | 137.79M | 0 | 93.01% | 69.02% |
| | SLAC-ANN | 0.46M | 137.80M | 0 | **93.24%** | **69.83%** |

performance. Through formulating the feature distributions, our proposed algorithm successfully alleviates the unfavorable gap between ANNs and CNNs. Compared to CNN, SLAC-ANN achieves competitive results on CIFAR-100 with only $0.44\%$ accuracy drop. The evaluation on ResNet-32 also demonstrates the superiority of our algorithm. Similarly, SLAC-ANN reduces the existing accuracy gap by $0.23\%$ on CIFAR-10 [$0.28\% \rightarrow 0.05\%$] compared to CNN and improves ANN by $0.81\%$ on CIFAR-100. We attribute this superiority to the elimination of existing overlapped areas of distribution tails, which boosts the potential classification performance of ANNs.

## 4.2 Experiments on ImageNet

We further conduct evaluation on ImageNet dataset [12], which contains $1.2M$ training images and $50k$ testing images with size of $224 \times 224$ from 1000 categories. The pre-processing and data augmentation follow the same protocols as in [9]. We make use of SGD optimizer with an initial learning rate of 0.1, weight decay of $1 \times 10^{-5}$, momentum of 0.9 and a cosine learning rate schedule. The entire training takes 300 epochs with a batch size of 256. The learning rate of trainable parameter $\Sigma$ is downscale by $1 \times 10^3$, $\lambda$ is set to 0.01 and $\beta$ to 0.1. The models are trained on 4 NVIDIA Tesla V100 GPUs. Similar to CIFAR experiments, we include CNN, ANN and BNN for comparison.

The results are reported in Table 2. We evaluate the performance of SLAC-ANN on ResNet-18 model. XNOR-Net replaces the multiplication by XNOR operations in ResNet trained on ImageNet [16]. Although high efficiency can be achieved, the performance gap is tremendous compared with CNN, with only $51.2\%$ top-1 accuracy and $73.2\%$ top-5 accuracy. On the contrary, ANNs can achieve competitive results. With proposed algorithm, we further narrow the existing gap between ANN and CNN. Comparing with vanilla ANN, SLAC-ANN improves the top-1 accuracy by $0.7\%$ [$67.0\% \rightarrow 67.7\%$] and top-5 accuracy by $0.3\%$ [$87.6\% \rightarrow 87.9\%$]. Similarly, we conduct experiments on ResNet-50. Comparing with vanilla ANN, SLAC-ANN improves the top-1 accuracy by $0.4\%$ [$74.9\% \rightarrow 75.3\%$] and top-5 accuracy by $0.9\%$ [$91.7\% \rightarrow 92.6\%$], which demonstrates that ANN classification performance can be improved through eliminating the overlapping tails.

## 4.3 Comparison with Loss Variants

To illustrate the necessity of the proposed mixture of multivariate skew Laplace distribution with angle-based constraint, we introduce several loss variant for comparison. We first include L-SoftMax

Table 2: Classification results on ImageNet datasets.

| Model | Method | #Mul. | #Add. | #XNOR. | Top-1 Acc | Top-5 Acc |
|---|---|---|---|---|---|---|
| ResNet-18 | CNN | 1.8G | 1.8G | 0 | 69.8% | 89.1% |
| | BNN | 0.1G | 1.8G | 1.7G | 51.2% | 73.2% |
| | ANN | 0.1G | 3.5G | 0 | 67.0% | 87.6% |
| | SLAC-ANN | 0.1G | 3.5G | 0 | **67.7%** | **87.9%** |
| ResNet-50 | CNN | 3.9G | 3.9G | 0 | 76.2% | 92.9% |
| | BNN | 0.1G | 3.9G | 3.8G | 55.8% | 78.4% |
| | ANN | 0.1G | 7.6G | 0 | 74.9% | 91.7% |
| | SLAC-ANN | 0.1G | 7.6G | 0 | **75.3%** | **92.6%** |

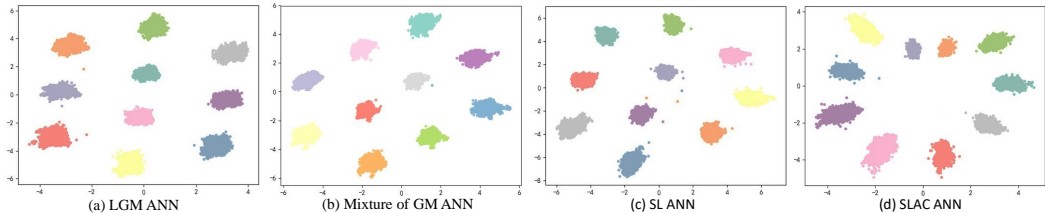

| (a) LGM ANN | (b) Mixture of GM ANN | (c) SL ANN | (d) SLAC ANN |

Figure 3: Visualization of feature distributions of LeNet on MNIST with various ANN variants. (a) denotes ANN with Large-margin Gaussian Mixture loss. (b) denotes ANN with mixture of Gaussian mixture. (c) denotes ANN with mixture of skew Laplace distributions. (d) denotes ANN with mixture of skew Laplace distributions equipped with angle-based constraint.

Table 3: Comparison of different ANN variants on CIFAR-10 and CIFAR-100 datasets.

| Classifier | L-SoftMax | LGM | MoM | SL | SLAC |
|---|---|---|---|---|---|
| CIFAR-10 | 91.83% | 92.00% | 91.95% | 91.99% | **92.29%** |
| CIFAR-100 | 67.53% | 67.56% | 67.73% | 67.51% | **68.31%** |

[14] as the baseline. We further compare SLAC-ANN with different types of distributions, such as the mixture of Gaussian distributions with large-margin (LGM) [22]. To further demonstrate the superiority of SLAC-ANN, we also include a modified version of LGM named a mixture of Gaussian mixture (MoM) as a stronger baseline which replaces the Gaussian distribution corresponding to each class in [22] with a mixture of two Gaussian distributions since the representation power of Gaussian mixture can be quite powerful to explore potential optimal distribution for ANNs. Skew Laplace distribution (SL) without angle-based constraint is also included for comparison.

First, we visualize the feature distributions of aforementioned baselines in Figure 3. Consistent with the analysis in Sec 3.1, the direction of distribution heavy-tailedness cannot be optimized with only lower-order moments $\mu$ and $\Sigma$, as shown in Figure 3 (a). We further visualize the feature distributions of MoM and SL ANNs, as shown in Figure 3 (b) and (c). In MoM, we replace the Gaussian distribution for each class with a mixture of Gaussian to explore larger space of distributions. The feature distribution tails of both MoM-ANN and SL-ANN form minor and irregular directions, which demonstrates that pre-defining the feature distributions alone cannot achieve desired properties that the heavy tails of different feature distributions form angle discrepancy. A visualization of proposed SL loss with angle-based constraint is shown in Figure 3 (d) where every class successfully forms a cluster with its distribution tail separated in various angle areas.

We further compare the performance of these ANN variants. All the variants are trained under the same training setting on CIFAR-10/100 with ResNet-20 mentioned in Sec 4.1. Note that all the baselines are trained with 800 epochs for a fair comparison. For the hyper-parameters of LGM and MoM, we adopt the same setting in [22]. The results are reported in Table 3. L-SoftMax can be treated as a naive approach for addressing the distribution overlapping issue through enlarging classification margin. However, L-SoftMax achieves the similar performance as vanilla ANN, which demonstrates that directly including margin optimization in SoftMax loss function could be ineffective in ANNs due to the heavy-tailedness. Comparing different distributions on CIFAR-10 dataset, ANN with LGM, MoM, and SL distribution achieve similar performance while SLAC-ANN achieves the best accuracy with 0.29% improvement compared to LGM. On CIFAR-100, the gaps become more obvious. Both LGM and SL cannot achieve competitive results. For LGM, we attribute this gap the natural limitation that single Gaussian distribution can hardly fit the long-tailed one in ANNs. For SL, although the heavy tails in ANNs can be easily fitted, the overlapped areas of them cannot be handled, which leads to misclassification. MoM achieves similar performance compared to vanilla ANN, which indicates the enlarged distribution space is not the major contributor to the improvement. SLAC achieves the best performance among all the baselines, which demonstrates its effectiveness.

## 4.4 Ablation Studies

**Number of Epochs.** As mentioned in Sec. 3.3, we enlarge the number of epochs due to the higher training complexity of SLAC-ANN. To demonstrate the effectiveness of proposed algorithm, we enlarge the training epochs to 800 for vanilla ANN and conduct experiments under different training epochs. We first compare the training and testing curves of SLAC-ANN and vanilla ANN. As

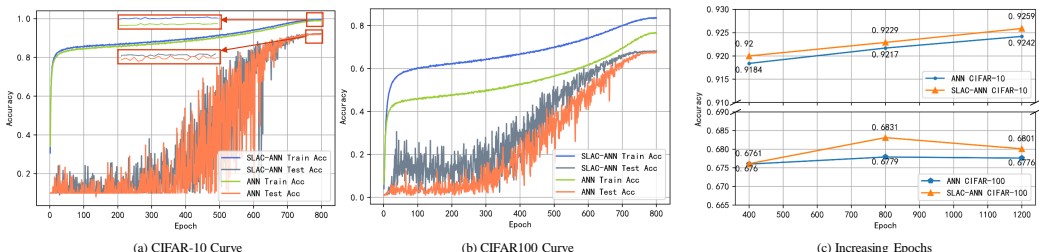

Figure 4: Training and testing accuracy curves with 800 training epochs on CIFAR-10 abd CIFAR-100 in (a) and (b) respectively. (c) denotes the testing accuracy of vanilla ANN and SLAC-ANN on CIFAR-10 and CIFAR-100 under different training epochs.

Table 4: Ablation Studies of SLAC-ANN on CIFAR-100.

|  | Location | Covariance | Skewness | $\lambda$ | $\beta$ | Accuracy |
|---|---|---|---|---|---|---|
| SL Parameters | ✗ | ✓ | ✓ | 0.01 | 0.1 | 64.82% |
|  | ✓ | ✗ | ✓ | 0.01 | 0.1 | 67.73% |
|  | ✓ | ✓ | ✗ | 0.01 | 0.1 | 62.04% |
| Loss Terms | ✓ | ✓ | ✓ | 0.0 | 0.0 | 67.39% |
|  | ✓ | ✓ | ✓ | 0.01 | 0.0 | 67.51% |
|  | ✓ | ✓ | ✓ | 0.0 | 0.1 | 67.73% |
| SLAC-ANN | ✓ | ✓ | ✓ | 0.01 | 0.1 | **68.31%** |

shown in Figure 4 (a), we train both SLAC-ANN and ANN with a ResNet-20 on CIFAR-10 where SLAC-ANN consistently has higher training accuracy than vanilla ANN, which enables SLAC-ANN to achieve better classification performance on CIFAR-10. The superiority becomes more obvious on CIFAR-100, as shown in Figure 4 (b). There exist large gaps between SLAC-ANN and vanilla ANN in terms of both training and testing curves, which empirically verifies that SLAC-ANN successfully improves the classification performance besides the enlargement of epochs. To further demonstrate the influence of training epochs, we conduct experiments on CIFAR-10 and CIFAR-100 under different training epochs including 400, 800 and 1200 epochs, as shown in Figure 4 (c). Through enlarging the training epochs on CIFAR-10, both SLAC-ANN and ANN achieve better performance, and SLAC-ANN consistently surpasses vanilla ANN with around 0.1% to 0.2% improvement under different epochs. On CIFAR-100, although SLAC-ANN has similar performance with ANN with 400 epochs, the superiority becomes more obvious with increasing epochs. We attribute this gap to the natural training complexity of SLAC which introduces a clustering algorithm with angle-based constraint on tails. After enlarge epochs to 800, SLAC-ANN achieves 0.52% accuracy improvements compared with ANN. When the epochs are enlarged to 1200, both SLAC-ANN and vanilla ANN reach sub-optimal areas with accuracy reduction. Thus, we set the training epochs to 800.

**Effectiveness of Different Components.** We conduct ablation studies of the proposed SLAC-ANN to verify the effectiveness of the distribution parameters and loss terms. Since we assume ANN features follow a mixture of Multivariate Skew Laplace distributions, all the distribution parameters including location, covariance, and skewness are studied. The influence of different loss terms is included in our studies. We make use of a ResNet-20 with ANN as the baseline model and evaluate all the variants on CIFAR-100. The results are reported in Table 4. For SL parameters, ✗ denotes corresponding parameters are fixed after initialization while ✓ denotes involved in optimization. As shown in rows of SL Parameters, all the distribution parameters contribute to the classification accuracy. Location and skewness play more importance roles since they are the key factors in clustering and proposed constraint while covariance models the variance of each feature dimension for different classes. Without the involvement of regularization term or angle-based constraint, all the variants have relatively large accuracy drop, as shown in the rows of Loss Terms. Through incorporating the optimization of all distribution parameters and including proposed loss terms, SLAC-ANN achieves the best performance.

## 5 Conclusion

Adder Neural Networks are more efficient for hardware designs and achieve a satisfactory energy reduction, which has wide application potential for computer vision tasks. To alleviate the existing performance gaps between ANNs and CNNs in classification task, we propose to investigate the potential natural weakness due to the replacement of operations. We argue that the heavy-tailed feature distributions in ANNs could lead to worse classification and propose to pre-define ANN features to follow a mixture of Multivariate Skew Laplace distributions, with which the heavy tails in ANNs can be better controlled with high order moment skewness. We introduce an angle-based constraint on distribution parameters to incorporate high diversity of distribution tails in angle space so that the overlapping can be eliminated. We conduct experiments on various models and datasets where the proposed SLAC-ANN consistently achieves superior performance.

## Acknowledgment

The authors would like to thank the Area Chair and the reviewers for their constructive comments. This work was supported in part by the Australian Research Council under Projects DE180101438 and DP210101859.

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
