# Handling Long-tailed Feature Distribution in AdderNets (Supplementary Material)

**Minjing Dong**[1,2], **Yunhe Wang**[2]*, **Xinghao Chen**[2], **Chang Xu**[1]
[1]School of Computer Science, University of Sydney
[2]Huawei Noah's Ark Lab
mdon0736@uni.sydney.edu.au, yunhe.wang@huawei.com,
xinghao.chen@huawei.com, c.xu@sydney.edu.au

## A    Hyper Parameters Studies

Besides the ablation studies reported in the paper, we further provide more results with different hyper parameters to demonstrate the influence of them. The results are shown in Table 1. Covariance column denotes the downscale rate of learning rate of parameter $\Sigma$. $\lambda$ and $\beta$ denote the hyper parameters of likelihood regularization and angle-based constraint respectively. As mentioned in the paper, location and skewness play more important roles than covariance in our empirical evaluations. However, the learning rate of covariance needs to be selected carefully. All the values in covariance parameters are defined positive and any negative value in $\Sigma$ leads to NaN during training. As show in the first two rows in 1, an appropriate downscale rate of learning rate for covariance can be set to $[0.001, 0.01]$ to obtain similar accuracy while $0.1$ leads to NaN value.

The performance of SLAC is sensitive to both $\lambda$ and $\beta$. Larger values of $\lambda$ and $\beta$ could easily overwhelming classification loss while smaller values could constrain their influence on the distribution parameters. As compared in Table 1, $\lambda$ is set to $0.01$ and $\beta$ to $0.1$ to obtain the best performance.

Table 1: Hyper Parameters Studies of SLAC-ANN on CIFAR-100.

| Covariance | $\lambda$ | $\beta$ | Accuracy |
|---|---|---|---|
| 0.1 | 0.01 | 0.1 | - |
| 0.001 | 0.01 | 0.1 | 68.11% |
| 0.01 | 0.001 | 0.1 | 67.81% |
| 0.01 | 0.1 | 0.1 | 65.76% |
| 0.01 | 0.01 | 0.01 | 67.83% |
| 0.01 | 0.01 | 1.0 | 67.63% |
| 0.01 | 0.01 | 0.1 | **68.31%** |

## B    SLAC on CNNs

To further demonstrate the generalization of proposed SLAC, we conduct experiments on CNNs. For a fair comparison, we adopt the same setting in Sec. 4.3 where all the models are trained for 800 epochs with a cosine learning rate schedule. We train vanilla CNN and SLAC-CNN with ResNet-20 on CIFAR-10 and CIFAR-100. The results are reported in Table 2. SLAC-CNN achieves slightly better performance than vanilla CNN with around $0.1\%$ accuracy improvement, which demonstrates that handling heavy tails in feature distribution could also benefit CNNs. Since ANNs have more obvious heavy-tailed problems based on our observations, SLAC improves ANN with a relatively

---

*Corresponding author.

35th Conference on Neural Information Processing Systems (NeurIPS 2021).

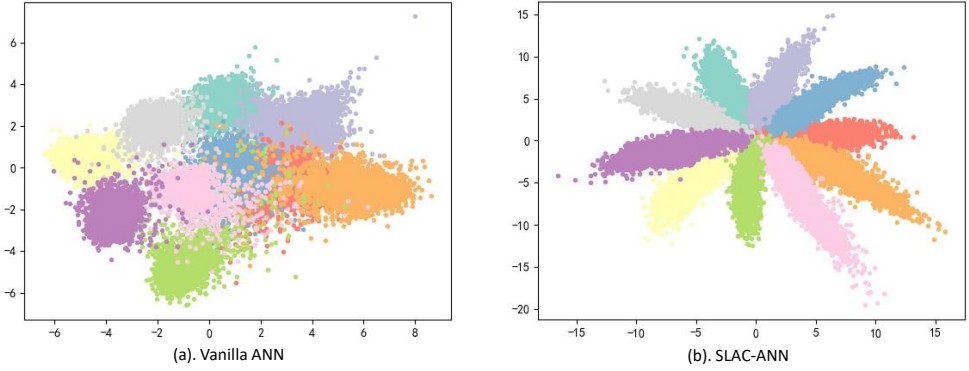

| (a). Vanilla ANN | (b). SLAC-ANN |

Figure 1: An illustration of proposed method. ANN feature extraction network remains the same while proposed classifier imposes constraints on features which involves both Mahalanobis and cosine distance to form a clustering classification with distribution tails dispersed in different directions.

large margin than CNN, which also empirically verified that addressing the overlapping of long-tailed feature distributions could be a key component in ANNs.

Table 2: Evaluation of SLAC on CNNs

| method | vanilla CNN | SLAC-CNN |
|---|---|---|
| CIFAR-10 | 93.44% | **93.54%** |
| CIFAR-100 | 70.22% | **70.31%** |

## C    Visualization of Long Tail Problem

The visualization in Figure 3 in Section 4.3 is a simple verification of our proposed SLAC on MNIST, which demonstrates that angle-based constraint can successfully achieve disentanglement in angle space. To further demonstrate the effectiveness of proposed SLAC-ANN and long tail problem, we conduct more visualization on CIFAR-10 with LeNet. As shown in Figure 1, vanilla ANN forms clustering based on $\ell_1$-norm, however, there exist obvious overlapping areas of different classes due to the long-tailed distributions of features. On the contrary, our proposed SLAC-ANN successfully achieve high diversity in angle space, which decreases the overlapped areas.