# OpenReview forum: "Handling Long-tailed Feature Distribution in AdderNets"
_NeurIPS.cc/2021/Conference — NeurIPS 2021 Poster_

### Official Review · Reviewer_aWB2 · 2021-07-13

**Rating:** 6
**Confidence:** 5

**Summary:**

This paper investigates the performance gap between ANNs and CNNs in classification task and focuses on the heavy-tailed feature distributions in ANNs. To alleviate this problem, this paper proposes to pre-define ANN features to follow a mixture of Multivariate Skew Laplace distributions, and introduces an angle-based constraint on distribution parameters to incorporate high diversity of distribution tails in angle space. Extensive experiments and ablation studies are conducted on CIFAR and ImageNet, which properly demonstrate the effectiveness of the proposed method.

**Limitations And Societal Impact:**

The limitations of adder neural networks are discussed. This paper improves the performance of adder neural networks, which are beneficial for energy-efficient neural networks.

**Main Review:**

This paper is well written and easy to understand. The motivation of dealing with long-tailed feature distribution of Adder neural network makes sense and is quite interesting. Based on the observations and analysis, this paper proposes to replace classifier head with appropriate trainable distribution parameters. Furthermore, an angle-based constraint is introduced for controlling feature distribution tails to avoid potential entanglement. The proposed method consistently improves the performance of AdderNets on CIFAR and ImageNet datasets. Extensive ablation studies are provided to justify the proposed method.

I have several concerns:
1. In Figure 1, could the feature distribution of CNN also be visualized to better demonstrate the long tails phenomenon of AdderNet. I wonder whether the long-tailed distribution also exists for CNN.
2. The proposed method shows good performance for ResNet-18 on ImageNet. I would like to see more experiments on ResNet-50, which will better demonstrate the effectiveness of the proposed method.

Minor comments:
1. L324, a angle-based -> an angle-based

---------------------------post rebuttal------------------------------
My concerns are addressed in the authors' rebuttal. I keep my score of 6.

**Time Spent Reviewing:**

3 hours

---

> ### Author Response · Authors · 2021-08-10
> **Response to Reviewer aWB2**
>
> Thanks for your constructive comments.
>
> **Long-tailed Distribution in CNNs**
>
> According to our empirical observation, long-tailed distribution also exists for CNNs. However, ANN features are more likely to follow long-tailed distributions and these long tails tend to be more critical than those in CNNs. This is because the prior of $\ell_1$ norm is Laplace distribution and ANNs make use of $\ell_1$ norm to measure the similarity of feature and filters. That is why ANN features tend to follow a long-tailed distribution instead of bell-curved one in CNNs. We have provided the results of applying SLAC to CNNs in supplementary material. SLAC-CNN only outperforms vanilla CNN slightly while SLAC-ANN boosts vanilla ANN.
>
> **Experiments on ResNet-50**
>
> We include the experiments of SLAC-ANN on ResNet-50 in the following table:
>
> | Method | #Mul. | #Add. | #XNOR. | Top-1 Acc | Top-5 Acc |
> | ------ | ------ | ------ | ------ | ------ | ------ |
> | CNN | 3.9G | 3.9G | 0   |76.2%|92.9%|
> | BNN | 0.1G | 3.9G | 3.8G |55.8%|78.4%|
> | ANN | 0.1G | 7.6G | 0   |74.9%|91.7%|
> | SLAC-ANN | 0.1G | 7.6G | 0   |75.3%|92.6%|

---

### Official Review · Reviewer_AHNP · 2021-07-14

**Rating:** 6
**Confidence:** 4

**Summary:**

The paper studies the potential influence due to different operations in similarity measurement, which could result in the worse classification performance of AdderNets compared with CNNs. Through empirical observations and toy examples, the paper focuses on the long-tailed feature distributions. The authors propose to formulate features to follow distributions with trainable parameters including skewness. A constraint is introduced to utilize sknewness to encourage tail divergence in angle space. Through extensive experiments, proposed SLAC-ANN outperforms all other baselines and achieves competitive results compared with CNNs.



**Limitations And Societal Impact:**

The long-tailed feature distribution could be only one of the factors which constrains the performance of ANNs. Other potential factors based on empirical observations or theoretical analysis remain unexplored and deserve more discussion.



**Main Review:**

This paper is organized with a clear motivation which prevents the existing long tails to be overlapped through pre-defining features to follow trainable skew Laplace mixture with skewness parameters for high-level control of feature tails. The proposed angle-based constraint seems reasonable, which establishes connection between cosine similarity and Mahalanobis distance. The authors verify the effectiveness of SLAC-ANN through sufficient evaluations under various scenarios. The improvement seems obvious.

Overall, the contribution is solid. My main concern lies in other simple baselines which can eliminate the long-tails feature distributions. Since the major risk comes from overlapped tails, why not apply some constraint to directly remove the long-tails, such as center loss? If these simple baselines can also solve this problem, what are the advantages of SLAC-ANN compared with them?

Minor Comments:
a fatter tails => fatter tails
an naive => a naive
be easily applied on => be easily applied to


**Time Spent Reviewing:**

8

---

> ### Author Response · Authors · 2021-08-10
> **Response to Reviewer AHNP**
>
> Thanks for your constructive comments.
>
> **Center loss**
>
> We agree that applying some popular constraints, such as center loss, could eliminate the long-tailed feature distribution in AdderNets. However, directly imposing these constraints always leads to worse results according to our empirical evaluations. For example, L-Softmax and LGM can eliminate long feature tails, however, both of them fail to outperform the baseline, as shown in Table 3. We also include center loss for comparison as suggested, which achieves 67.66%, with 0.65 reduction compared to SLAC. We attribute this to natural property of ANNs where the prior of $\ell_1$-norm is Laplace distribution which has long tails, as discussed in Line 45-47. Thus, instead of elimination of long tails, SLAC imposes angle-space constraint to encourage diversity of feature tails in angle space.
>
> **Minor Comments**
>
> Thanks for your suggestions. These flaws will be fixed in the final version.

---

> > ### Comment · Reviewer_AHNP · 2021-08-22
> > **Thanks for the feedback**
> >
> > I think my concerns have been addressed. So I keep my score as Maginally above the borderline.

---

### Official Review · Reviewer_P2Ke · 2021-07-14

**Rating:** 6
**Confidence:** 4

**Summary:**

The paper proposes to alleviate the gap between CNNs and ANNs through addressing the heavy-tailed features in ANNs. The authors argue that the observed heavy tails in feature distributions can be easily overlapped, which limits the classification performance of ANNs. To remove this risk, the skew Laplace distribution with higher-order moment is introduced to be embedded into loss function through involving likelihood regularization and conditional probability computation. With introduced distributions, the angle-based constraint is proposed for arranging for long tails of different classes to lie in their corresponding angle regions so that the overlapping of tails can be avoided.

**Limitations And Societal Impact:**

Proposed algorithm is designed for ANNs and might lack generalization to other networks, such as CNNs. The authors discuss this limitations in supplementary material.


**Main Review:**

Strength:
+ The paper is well-organized and easy to follow, with motivation well-explained.
+ The proposed algorithm is interesting and novel. The involvement of higher-order moment seems natural for directly constraining the heavy tails. The introduced constraint is also explicit, which imposes high diversity of tails in angle space.
+ Extensive empirical evaluations on different architectures and datasets. Superior performance compared to other baselines. The feature visualization demonstrates that proposed algorithm can successfully impose angle-based constraint. Ablation studies verifies the effectiveness of each components.

There exist several concerns:
- The classifier in ANN is replaced by the distribution parameters, including location, covariance and skewness. The classifier forwarding is replaced by the distribution density function in Eq. 2. However, any increment of the number of parameters or FLOPs is not discussed in this paper.
- In Eq. 8, the authors introduce a target distance \hat{D_{gamma}} which makes use of the current Mahalanobis distance between center locations among classes to derive an adaptive target. However, the reason of it is not well-explained. I wonder how the angle-based constraint performs when the target distance is replaced by a fixed one with equal intervals in range [-1, 1]? It seems that a fixed target can also meet the requirements that avoids overlapped tails.


**Time Spent Reviewing:**

Three hours.

---

> ### Author Response · Authors · 2021-08-10
> **Response to Reviewer P2Ke**
>
> Thanks for your constructive comments.
>
> **Increment of FLOPS**
>
> As shown in Figure 2, the only difference lies in the classifier where SLAC replace the linear layer in vanilla ANN with SL parameters. Since SL parameters contains three parameters including location, covariance and skewness, the parameters and FLOPs of classifier are increased accordingly. However, note that the number of parameter and FLOPs of classifier are much smaller than those in feature extractor network, which is nearly ignorable, as calculated in Table 1 and 2.
>
> **Fixed Intervals for Angle-based Constraint**
>
> As illustrated in Figure 2, there exists a strong correlation between the distances and angles among class centers. Through establishing such a connection in Eq. 8, all the SL parameters are optimized by the angle-based constraint where the location parameters are also optimized to have larger diversity. If a fixed adaptive target distance is used, only the skewness parameter in SL distribution will be optimized by the angle-based constraint, which limits the influence of SLAC loss. To evaluate this variant, we also include the experiment which replaces adaptive target distance with a fixed one with equal intervals on CIFAR100 with ResNet20. And this baseline achieves 67.74%, with a relatively large reduction compared with SLAC-ANN which achieves 68.31%.

---

### Official Review · Reviewer_rrae · 2021-07-16

**Rating:** 7
**Confidence:** 5

**Summary:**

This paper presents a potential problem in AdderNets where the long-tailed features could be difficult for classification tasks owing to the entanglement of feature tails. To achieve disentanglement, the authors assume features of each class follow Skew Laplace (SL) distribution to obtain better control of feature tails. With the skewness parameters in SL distribution, the authors propose to eliminate overlapped areas through imposing constraints on the directions of feature tails. The entire algorithm replaces the classifier with distribution parameters and can be updated by SGD. Experiments are conducted on several datasets and models to demonstrate the advantage of the proposed algorithm.

**Ethical Concerns:**

No ethical concern

**Limitations And Societal Impact:**

Well addressed in this paper and I have no particular concern

**Main Review:**

The presentation of this paper is clear. The authors demonstrate a novel approach for addressing long-tailed feature distributions in AdderNets. The assumption seems reasonable and the reasons why SL distributions are needed in AdderNets are well-justified. The experimental results seem promising and sufficient enough. The comparison on various datasets shows strong evidence that the proposed algorithm can address the existing problem and improve the performance of AdderNets. The ablation studies show the influence of each loss term and distribution parameters.

My major concern is the selection of distance in Eq. 7, which uses Mahalanobis distance for measuring the location difference among class centers. However, there is no explanation for this choice. And there is no other simple baseline comparison in ablation studies, such as Euclidean distance.

Some minor points:
1. AdderNet and ANN are mixed.
2. Multiple citations for one paper [11, 12], [29,30].
3. Miss relevant citations of Skew Laplace distributions.


**Time Spent Reviewing:**

3

---

> ### Author Response · Authors · 2021-08-10
> **Response to Reviewer rrae**
>
> Thanks for your constructive comments.
>
> **Selection of Distance in Eq. 7**
>
> SLAC makes use of Mahalanobis distance for measuring the distance among class centers, which naturally comes from the probability density function of SL distribution, as shown in Eq. 2 where the distance between feature and class centers are computed by Mahalanobis distance. Actually, Mahalanobis distance reduces to Euclidean distance when the covariance matrix is the identity matrix. For a precise measurement of distances among class centers, Mahalanobis distance is used in SLAC. To verify this statement, we also conduct experiment which replaces Mahalanobis distance with Euclidean distance in SLAC and redo the experiment on CIFAR-100 with ResNet20. The one with Euclidean distance achieves 68.11%, with 0.2% reduction compared with Mahalanobis distance (68.31% to 68.11%).
>
> **Minor points**
>
> Thanks for your suggestions. These flaws will be fixed in the final version.

---

### Decision · Program_Chairs · 2021-09-28

**Decision:**

Accept (Poster)

**Comment:**

The manuscript has been reviewed by four experienced reviewers, all of whom acknowledged the contributions of the submission and recommend acceptance. Specifically, the reviewers agree that the manuscript is easy to follow and the proposed method is novel.

Since there is no attempt to reject the submission, there is no basis to overturn the consensus. The AC thus recommends an acceptance.

**Consistency Experiment:**

NeurIPS has a long history of experimentation. In 2014, NeurIPS ran an experiment in which 10% of submissions were reviewed by two independent committees to quantify the randomness in the review process. This year, we repeated a variant of this experiment to see how the quality of the review process has changed over time.  This paper was part of the experiment and was therefore assigned to two committees (consisting of reviewers, an Area Chair, and a Senior Area Chair) that reached independent decisions.  If both committees made the same recommendation, this recommendation was followed. If a single committee recommended acceptance, the paper was accepted (with the exception of a few cases in which the other committee identified what we considered a fatal flaw, e.g., an error in a key result).

This copy’s committee reached the following decision: **Accept (Poster)**

The other committee assigned to the paper recommended **Reject**.  You can find the other set of reviews, along with any follow up discussion with the authors here:
https://openreview.net/forum?id=io4oeP0W5TA